# Innovative Application of High-Precision Seismic Interpretation Technology in Coalbed Methane Exploration

Chunlei Li [1], Lijiang Duan [1,*], Xidong Wang [2], Xiuqin Lu [3], Ze Deng [1] and Liyong Fan [4]

[1] PetroChina Research Institute of Petroleum Exploration & Development, Beijing 100083, China; lichunlei@petrochina.com.cn (C.L.); dengze@petrochina.com.cn (Z.D.)
[2] School of Energy, China University of Geosciences, Beijing 100083, China; 18232462327@163.com
[3] Exploration and Development Research Institute of HuaBei Oilfield Company, Renqiu 062550, China; yjy_lxq@petrochina.com.cn
[4] Research Institute of Exploration and Development, PetroChina Changqing Oilfield Company, Xi'an 710018, China; lyfan123_cq@petrochina.com.cn
* Correspondence: duanlj@petrochina.com.cn

## Abstract

Exploration of coalbed methane (CBM) has long been plagued by critical technical challenges, including a low signal-to-noise (S/N) ratio in seismic data, difficulty identifying thin coal seams, and inadequate accuracy in interpreting complex structures. This study presents an innovative methodological framework that integrates artificial intelligence (AI) with advanced seismic processing and interpretation techniques. Its effectiveness is verified through a case study in the North Bowen Basin, Australia. A multi-scale seismic data enhancement approach combining dynamic balancing and blue filtering significantly improved data quality, increasing the S/N ratio by 53%. Using deep learning-driven, multi-attribute fusion analysis, we achieved a prediction error of less than $\pm 1$ m for the thickness of thin coal seams (4–7 m thick). Integrating 3D coherence and ant-tracking techniques improved the accuracy of fault identification, increasing the fault recognition rate by 30% and reducing the spatial localization error to below 3%. Additionally, a finely tuned, spatially variable velocity model limited the depth conversion error to 0.5%. Validation using horizontal well trajectories revealed that the rate of reservoir encounters exceeded 95%, with initial gas production in the predicted sweet spots zone being 25–30% higher than with traditional methods. Notably, this study established a quantitative model linking structural curvature to fracture intensity, providing a robust scientific basis for accurately predicting CBM sweet spots.

**Keywords:** coalbed methane; high precision seismic interpretation; artificial intelligence; seismic attributes; geological modeling; sweet spot prediction

## 1. Introduction

Coalbed methane (CBM) is a typical unconventional natural gas resource characterized by thin coal seams, rapid lateral variability, and structurally complex reservoirs. Conventional seismic techniques are significantly limited in their ability to resolve thin seams and subtle structural features, which hinders the efficiency of CBM exploration and development [1,2]. Accurately delineating the spatial distribution of coal seams, their thickness variations, and associated small-scale faults is crucial for detailed reservoir characterization and efficient CBM production [3]. However, achieving high-resolution imaging of thin coal seams in shallow, noisy environments remains a longstanding challenge.

In recent years, a series of seismic processing and interpretation techniques tailored to CBM have been proposed to address these challenges. Traditional seismic data processing methods, such as, stacking, band-pass filtering, and amplitude scaling, can improve data quality to some extent, but they are often insufficient for resolving thin seams and fine-scale structures in CBM settings [4]. For example, to improve vertical resolution and signal clarity, Taner and Koehler (1981) proposed the dynamic balancing technique, which enhances weak reflection signals by dynamically adjusting the amplitude distribution of seismic traces [5]. Similarly, Partyka et al. (1999) introduced blue filtering to emphasize high-frequency components in seismic records, significantly improving the resolution of thin-bed reflections [6]. Despite these contributions, the application of these methods in shallow, structurally complex CBM environments is still constrained. In practice, it has been difficult to simultaneously achieve high resolution and a high signal-to-noise (S/N) ratio using conventional processing alone, as boosting high frequencies often also amplifies noise.

Seismic attribute analysis has become an important tool for improving CBM reservoir prediction accuracy [7]. Curvature and coherence attributes, proposed by Roberts (2001) and Chopra and Marfurt (2007), respectively, enhance the detection of small-scale features such as minor faults and fractures and are now widely used in conventional interpretation workflows [8,9]. However, traditional single-attribute analysis often neglects the rich nonlinear information embedded in seismic data, resulting in suboptimal prediction accuracy [10]. To address this, Marroquín et al. (2009) [11] conducted multi-attribute fusion for coal seam thickness prediction in the San Juan Basin, achieving greater accuracy through attribute integration. Nevertheless, most fusion techniques have relied on basic linear or statistical models and lack the capacity to fully exploit complex, nonlinear data relationships, often falling short of the desired predictive performance [11].

Recently, artificial intelligence (AI)-based methods have been increasingly applied in seismic interpretation. For example, Gao et al. (2022) proposed a multiscale convolutional neural network that greatly improved the accuracy and reliability of fault detection in seismic images [12]. However, the current application of AI in the CBM domain is still in its infancy, often limited by poor data quality and a lack of geological specificity in the models (e.g., models trained on conventional reservoirs may not generalize well to CBM) [13]. Therefore, effectively integrating AI with high-precision seismic interpretation techniques to improve thin-seam detection and structural interpretation remains a critical research challenge in CBM exploration.

In summary, the specific technical problems addressed by this study are: (i) the low detectability of thin coal seams under noisy conditions, (ii) the inaccuracy in mapping fine-scale faults and structural discontinuities, and (iii) the difficulty in predicting sweet spots (high-yield zones) in complex CBM reservoirs. Previous studies have made progress on individual aspects of these problems, but significant gaps remain. For instance, high-frequency boosting and amplitude balancing improved resolution yet often at the expense of noise amplification, and conventional multi-attribute analyses improved predictions but did not capture nonlinear correlations in the data. Moreover, while early applications of machine learning in seismic interpretation show promise, they have not been fully leveraged for CBM due to data limitations [12,13]. These gaps highlight the need for a more comprehensive and innovative approach. Additionally, recent international studies, e.g., Int. J. Coal Geology 2020–2024; Energy Reports 2021–2023, have applied AI-driven attribute fusion and thin-bed prediction, but their applicability to CBM remains limited.

In order to overcome the above bottlenecks, this study proposes a novel high-precision seismic interpretation framework that synergistically combines advanced processing, attribute analysis, and AI. Unlike previous approaches that apply a single technique in

isolation, our framework integrates dynamic balancing and blue filtering for multi-scale seismic enhancement, multi-attribute fusion aided by deep learning for thin seam thickness prediction, and advanced 3D coherence–curvature–ant tracking for automatic fault extraction. The motivation for this integrated approach is to simultaneously improve data resolution and continuity, capture complex geological patterns, and leverage data-driven algorithms to enhance prediction accuracy. By addressing thin seam imaging and subtle fault detection together, and by quantitatively linking structural curvature to fracture intensity for the first time, our method aims to provide a more reliable basis for identifying CBM sweet spots. In this paper, we demonstrate the application of the proposed framework in a CBM field in the North Bowen Basin of Australia, showing how it accurately predicts coal seam thickness and fault structures, and establishing a new quantitative relationship between structural curvature and fracture development. The results illustrate a significant improvement in reservoir characterization precision and offer a new technological paradigm and theoretical basis for efficient exploration and development of structurally complex CBM fields.

In contrast to previous studies that primarily applied isolated techniques, our framework provides a systematic integration of multi-scale processing, attribute fusion, and AI, which has not been previously demonstrated in CBM exploration.

## 2. Overview of the Study Area

The study area, which is part of a CBM field, is located on the north edge of the North Bowen Basin in Queensland, Australia. The North Bowen Basin is a rift basin that developed into a back-arc foreland basin on the basement of Early Paleozoic metamorphic and sedimentary rocks. It trends north–south, with a length of about 1100 km, a maximum width of 300 km, and an area of $1.6 \times 10^5$ km$^2$. The study area covers an area of approximately 72.7 km$^2$, including 5 seismic surveys (Figure 1). It has been extensively drilled with 293 wells in total, forming a dense network of wells. Coal maturity is moderate (Ro: 1.0–1.9%), and current development primarily targets the Rangal, Fort Cooper and Moranban coal measures. The Rangal coal measures comprise four coal seams (LU, LL, VU and VL). The Fort Cooper coal measures contain seven seams (Girrah, FCCM5, FCCM4, FCCM3, FCCM2, FCCM1 and Fairhill), which are generally characterized by thick, interbedded coal and mudstone with high ash content and poor coal quality. The Moranbah coal measures include five main seams (Q, P, GM, GL and DL), with GM being the primary target for development [14,15].

Structurally, small-scale normal faults are prevalent in the study area. While individual faults have relatively small displacements (typically less than 20 m), their dense distribution significantly impacts the continuity of coal seams and the connectivity of gas reservoirs. The thickness of the coal seams varies greatly across the study area. For example, the Fairhill seam is one of the thickest and most laterally continuous (50–100 m thick), whereas seams such as Q, GM and GL are much thinner, measuring just a few to tens of meters, and they exhibit rapid lateral variation. Seams such as Q, GM and GL are much thinner, measuring just a few to tens of meters, and they exhibit rapid lateral variation. In some western sub-regions, local erosion or non-deposition has completely removed certain thin seams. This high variability in seam thickness, combined with complex faulting, poses significant challenges for exploration and development. It necessitates the use of high-resolution, high-precision seismic interpretation technologies for detailed characterization.

Given the geological setting and data availability, this study area provides an ideal testing ground for our high-precision interpretation framework. The dense well control and high-quality 3D seismic data allow us to rigorously evaluate the performance of advanced techniques. In particular, we focus on addressing three key challenges in this

area: accurately mapping the thickness of thin coal seams, interpreting fine-scale faults, and predicting CBM sweet spots, i.e., areas of likely higher productivity. The following sections detail the methodology we developed and its application to surmount these challenges.

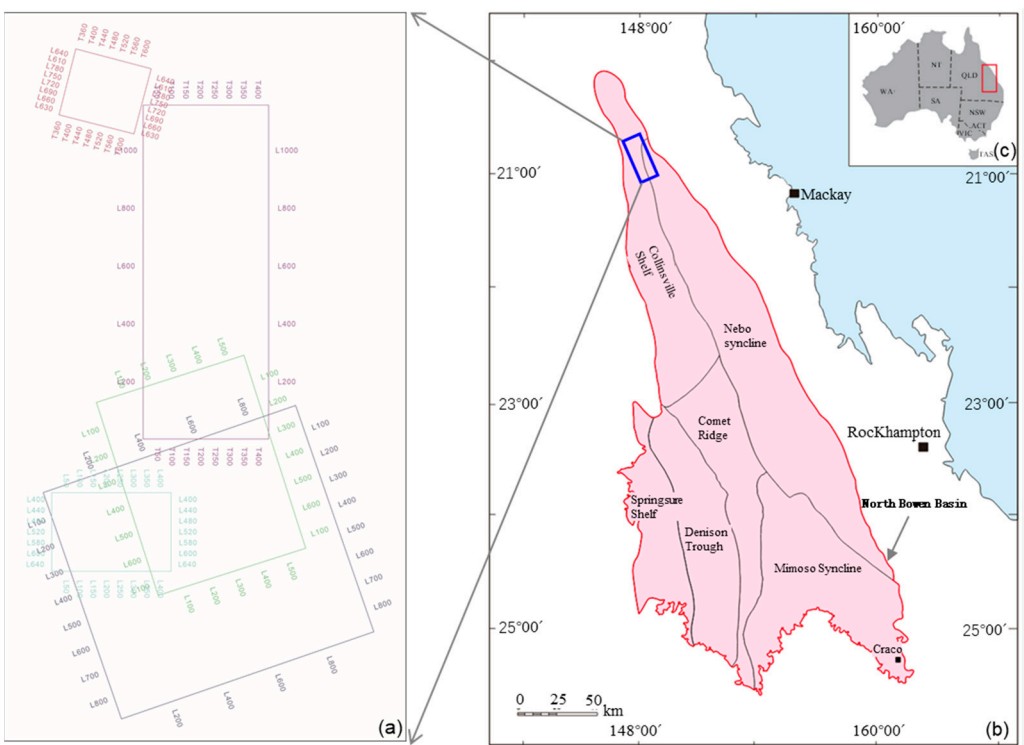

**Figure 1.** Study area (**a**) in North Bowen Basin (**b**) of Australia (**c**).

## 3. Methodology

In order to tackle the difficulties of identifying thin coal seams and subtle faults in CBM exploration, we have developed a comprehensive high-precision seismic interpretation workflow (Figure 2). This workflow integrates seismic data enhancement, multi-attribute analysis, and AI-based prediction. The overall process consists of the following key steps: (a) seismic data optimization (noise attenuation and resolution enhancement), (b) seismic attribute extraction and modeling, (c) automated structural feature recognition (fault and fracture extraction), (d) multi-volume data integration (merging multiple surveys), (e) spatially variable velocity modeling and time-depth conversion, and (f) 3D geological model construction. In this section, we describe the core methodologies and theoretical principles underlying these steps, with an emphasis on the critical procedures and parameters that ensure the approach is reproducible.

All experiments were conducted on a workstation with Intel Xeon CPUs, 128 GB RAM, and an NVIDIA RTX A6000 GPU (Dell Technologies, Inc., Round Rock, TX, USA), using Petrel 2022, GeophPro 2023, and Python 3.9 with TensorFlow 2.8. These specifications ensure reproducibility of the workflow.

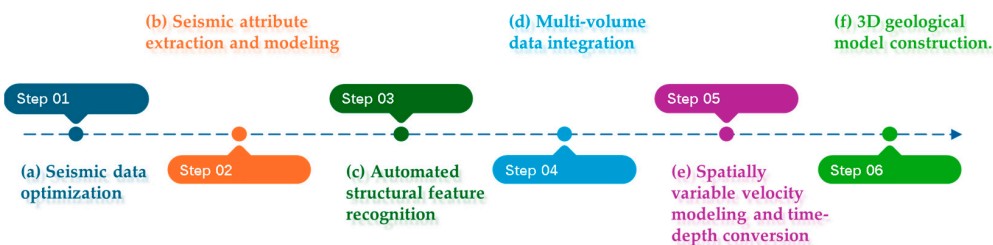

**Figure 2.** Workflow of the framework (**a–f**).

### 3.1. Dynamic Balancing of Seismic Amplitudes

Dynamic balancing is a post-stack amplitude enhancement technique, also known as automatic gain control, that strengthens weak reflection energy and improves the overall amplitude balance along the seismic trace. It works by suppressing overly strong amplitudes while amplifying weaker signals, dynamically adjusting the amplitude distribution of seismic traces. This improves the imaging quality of low-reflectivity targets, such as thin coal seams, by bringing out subtle reflections that would otherwise be obscured by stronger events or noise [7]. In our CBM dataset, shallow coal seam reflections are often masked by strong background noise and near-surface reverberations. The application of dynamic balancing markedly improved the visibility and continuity of these weak reflectors.

We implemented dynamic balancing through a moving window normalization approach. For a given seismic trace $x(t)$, a time-varying gain function $G(t)$ is computed as the root-mean-square (RMS) amplitude within a sliding time window of length $T$ centered at time $t$:

$$G(t) = \sqrt{\frac{1}{T} \int_{t-T/2}^{t+T/2} [x(\tau)]^2 d\tau}, \tag{1}$$

where $T$ is the window length (a key parameter) and $\tau$ is the integration variable. In this study, we chose a window length of 200 ms, which is on the order of the dominant period of the target reflections, to balance responsiveness and stability of the gain. The dynamically balanced trace $y(t)$ is obtained by dividing the original signal by the local gain with a small regularization constant $\epsilon$ to avoid division by zero:

$$y(t) = \frac{x(t)}{G(t) + \epsilon} \tag{2}$$

This process amplifies weaker signal portions (where $G(t)$ is small) and attenuates excessively strong portions (where $G(t)$ is large), yielding a more uniform amplitude distribution. We applied this normalization iteratively in areas with particularly strong amplitude contrasts until the trace amplitudes reached a balanced state. Figure 3 illustrates the effect of dynamic balancing on a synthetic seismic trace: the orange curve is the original trace, and the dark blue curve is after applying sliding-window normalization, which boosts weak reflector amplitudes and suppresses overly strong signals.

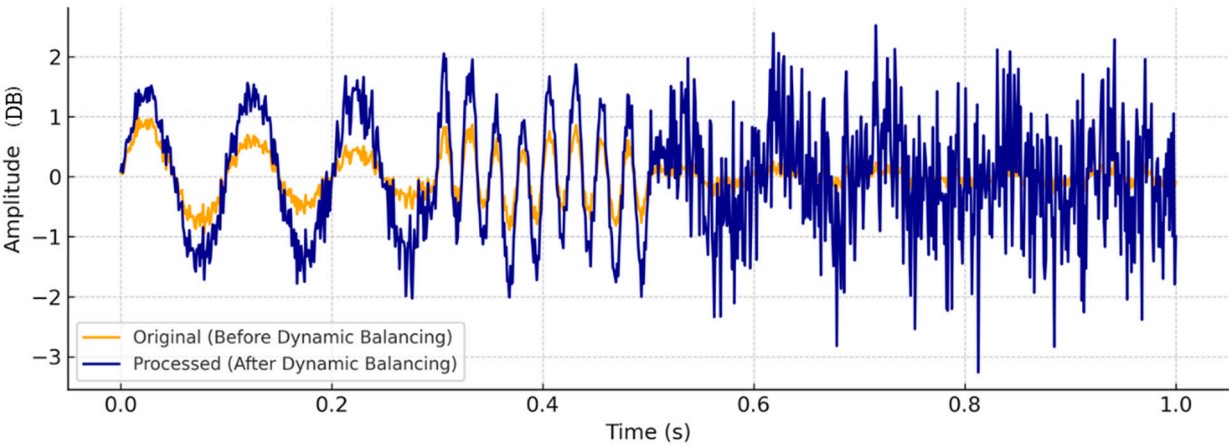

**Figure 3.** Effectiveness of dynamic equilibrium processing on simulated seismic traces.

### 3.2. Post-Stack Blue Filtering

Blue filtering is a spectral balancing technique that accentuates the high-frequency content of seismic data by reshaping its amplitude spectrum to resemble a 'blue' spectrum,

which is rich in high frequencies. This process selectively suppresses low-frequency energy while amplifying high-frequency components, thereby making the output spectrum closer to that of true reflection coefficients [6,16]. By compensating for frequency-dependent attenuation and imprinting an inverse of the typical red (low-frequency dominated) spectrum of seismic data, blue filtering can dramatically improve the resolution of thin beds. In our context, it enhances the reflectivity and detectability of thin coal seams and makes small fault terminations more visible. We found that applying blue filtering in combination with dynamic balancing is particularly effective: the former increases resolution, while the latter ensures those newly enhanced high-frequency reflections are not overwhelmed by residual noise, thus jointly improving both S/N ratio and vertical resolution.

Implementation and Parameters: Our blue filtering implementation involved three main steps:

(1) Extract Target Spectrum: We first derived a target high-frequency-rich spectral shape from well-log-derived reflectivity or synthetic seismic models. This target spectrum represents the desired "whitened" or "blue" frequency content, emphasizing frequencies above ~30 Hz that are crucial for thin bed resolution in this area.

(2) Design Filter Operator: We designed a spectral balancing operator that, when applied to the original seismic data, boosts the high frequencies to match the target spectrum. Practically, this operator is a frequency-domain gain function that raises the amplitude of higher frequencies, e.g., using a gentle ramp that increases with frequency, while damping low frequencies. The operator was tapered to avoid introducing sharp spectral artifacts and was constrained to unity gain at the dominant frequency to preserve overall amplitude levels.

(3) Apply Filtering: The operator was convolved with the seismic traces (equivalently, multiplication in the frequency domain) to produce the frequency-enhanced output data.

Key parameters in this process include the frequency range of interest (we focused on the 10–80 Hz band of our data) and the shape of the boost function. We chose a boost that gradually increased gain up to about +12 dB at 60 Hz (relative to 20 Hz baseline), based on observed spectral decay in the raw data. This achieved a balance between amplification of useful signal and control of noise. This is an example of a blue filter operator design derived from well log reflectivity (Figure 4). It shows higher gains at high frequencies ('blue' spectrum shaping). Figure 5 shows the effect of the blue filtering technique on seismic traces (synthetic example). The filtered trace shows enriched high-frequency content and clearer depiction of a thin seam reflection and a minor fault termination, compared to the original trace.

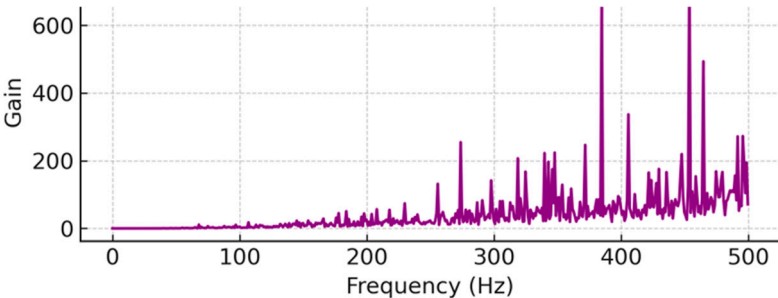

**Figure 4.** Blue filter operator.

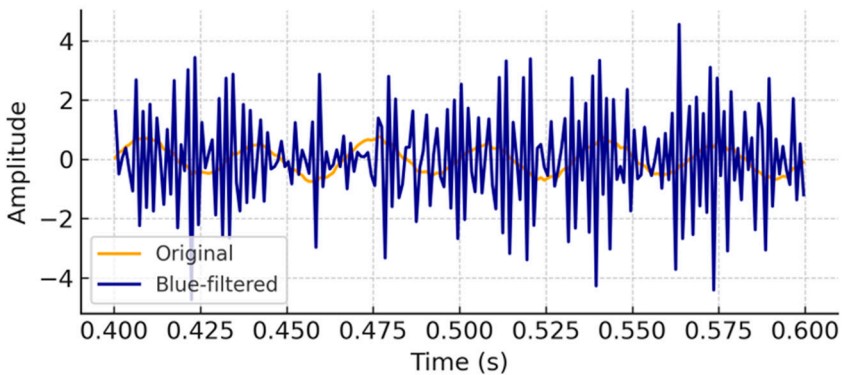

**Figure 5.** Effectiveness of blue filtering technique on simulated seismic traces.

*3.3. Seismic Attribute Analysis*

Seismic attributes are quantitative parameters extracted from seismic data that reflect specific geological features. These include amplitude, frequency, phase and derived volumes. In CBM reservoir characterization, coherence attributes help to identify discontinuities, such as faults and fracture zones. Curvature attributes quantify layer bending and highlight subtle structural features, such as minor folds and faults. Amplitude-based attributes, such as RMS amplitude and relative acoustic impedance, indicate coal seam thickness and lithological contrast. Spectral attributes, such as instantaneous and dominant frequency, help to identify the tuning effects of thin seams [17–21].

In terms of the specific technical implementation, this study uses the 3D grid interpolation technique to convert spatially discrete seismic amplitude data points ($x_i$, $y_i$, $z_i$ and $u_i$) into continuous 3D data by constructing an objective function with curvature constraints and a principle of minimizing the fitting error. This process is optimized by constructing an objective function where the first term ensures the interpolated surface is smooth through an energy minimization principle (e.g., curvature constraints) and the second term ensures the interpolated results minimize the error in fitting to the known data points. The optimization process involves solving the corresponding nonlinear partial differential equations iteratively to obtain the interpolating function across the entire 3D space.

$$A = \int E \, d\Omega + \sum_{i=1}^{n} [Mf(x_i, y_i, z_i) - u_i]^2 \tag{3}$$

Nonlinear partial differential equation:

$$F\left(x, y, \frac{\partial u}{\partial x}, \frac{\partial u}{\partial y}, \frac{\partial^2 u}{\partial x^2}, \frac{\partial^2 u}{\partial y^2}, \frac{\partial^2 u}{\partial x \partial y}, k\right) = 0 \tag{4}$$

Equations (3) and (4) describe the mathematical foundation of the interpolation method used in this study. Specifically, Equation (3) defines an optimization functional that combines two competing requirements: surface smoothness and data fidelity. The first term of the functional, expressed as an energy integral, imposes curvature constraints on the interpolated function and ensures that the resulting surface remains smooth and geologically reasonable. The second term represents the sum of squared errors between the interpolated values and the known data points $(x_i, y_i, z_i, u_i)$, thereby enforcing consistency with observed seismic attribute values. Minimization of this functional balances the trade-off between honoring the data at control points and maintaining overall smoothness in three dimensions. Equation (4) is the Euler–Lagrange equation derived from the optimization functional. It is a nonlinear partial differential equation involving first- and second-order derivatives of the interpolated function. This equation characterizes the necessary condition

for minimizing the functional and, consequently, governs the spatial distribution of the interpolated seismic attribute field. The curvature-related terms in the equation act as a regularization mechanism to suppress unrealistic oscillations, while the data-fitting terms enforce convergence towards the measured values. Because of its nonlinear nature, the equation is solved iteratively using finite-difference or finite-element methods until the interpolated solution converges to a stable state across the entire 3D volume.

In essence, Equation (3) establishes the optimization criterion for 3D interpolation, whereas Equation (4) provides the governing differential equation that ensures the interpolated seismic attribute field is both smooth and consistent with observed data. Together, they form the mathematical basis for constructing reliable 3D models of coal seam thickness and related reservoir properties in CBM exploration.

After constructing such volumes, we further enhanced interpretability by applying AI-assisted color palette mapping and minimum-curvature trend analysis. In practice, this involved using unsupervised machine learning (clustering) to automatically distinguish between "high", "medium" and "low" thickness zones, for example, and to color-code attribute values into meaningful categories. Curvature attributes were also used to guide trend interpretation, for instance by highlighting trends along fold axes or fault corridors. Additionally, we generated thin horizontal "slice" maps of amplitude or other attributes along interpreted seam horizons. In these maps, warmer colors indicate higher values, e.g., thicker seams or higher gas indicators, and cooler colors indicate lower values, with abrupt spatial changes often highlighting faults or deformation zones that disrupt the continuity.

The key innovation in our attribute analysis lies in the fusion of multiple attributes to predict reservoir properties, particularly coal seam thickness and fracture density, using AI algorithms. Instead of relying on visual overlay or simple linear regressions, we utilized a data-driven approach.

We assembled a training dataset from well locations where the target property (seam thickness, in meters) was known from logs. For each such location, we extracted the corresponding seismic attribute values, such as amplitude, frequency, and curvature, from the volumes described above. Then we trained a machine learning model to learn the relationship between the seismic attributes (input features) and the known target property. We experimented with both linear multi-attribute regression and nonlinear models. The best results were obtained using a feed-forward neural network (a type of deep learning) with a single hidden layer containing 12 neurons and nonlinear activation. This network was trained using 70% of the well data for training and 30% for validation, employing early stopping to avoid overfitting. The model effectively learned to combine attributes, such as a certain weighted mix of amplitude and frequency attributes, to predict seam thickness. After training, the model was applied to the entire seismic volume to compute a predicted thickness at every trace location, resulting in a 3D volume (or horizon map) of coal seam thickness.

The multi-attribute fusion approach is similar in spirit to the methodology of Hampson et al. (2001) [21,22], but here we enhanced it by including nonlinear learning (i.e., the neural network) to capture complex relationships, as well as integrating attributes that incorporate our advanced processing (i.e., dynamic-balancing-adjusted amplitudes) and geometric features (e.g., curvature and coherence) which are particularly relevant in a fractured, thin-bedded setting.

We withheld a subset of wells entirely from the modeling process and later compared the predicted thickness at those locations to the actual values. The prediction error for the target GM coal seam was within $\pm 1$ m for the vast majority of blind wells, which is an excellent accuracy given that the seam thickness ranges only 4–7 m. Figure 6 presents a representative plan-view map of the multi-attribute fusion result (with AI assistance) for

seam thickness, clearly delineating the spatial distribution of thicker coal sweet spots. The trends of thicker zones and their alignment or termination against faults (as interpreted from curvature and coherence) are evident, giving confidence that the fusion model is capturing real geological features and not just noise.

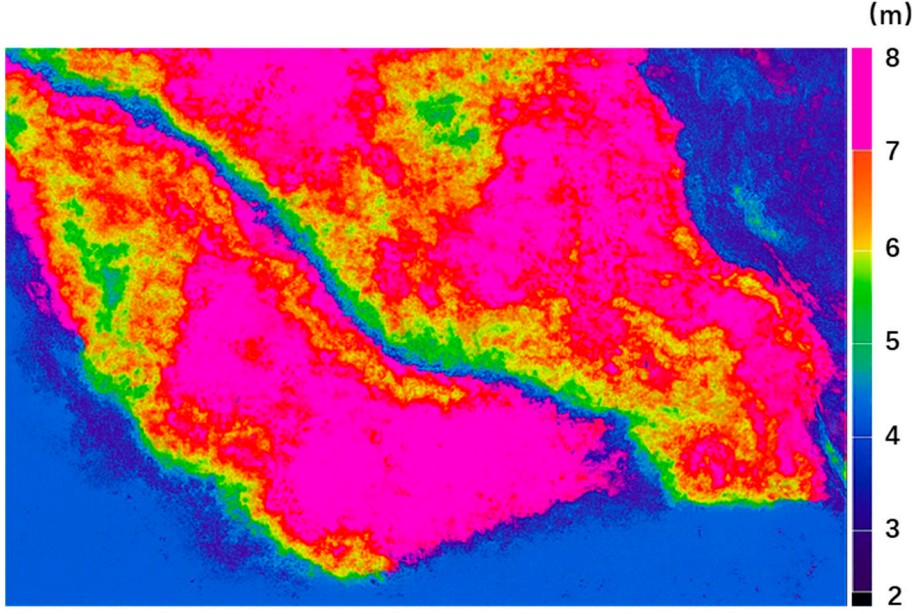

**Figure 6.** Multi-attribute joint analysis and AI-assisted interpretation application effect floor plan.

Planar map of multi-attribute fusion results with AI-assisted interpretation, highlighting coal seam sweet spot trends. Warmer colors indicate areas predicted to have thicker coal (and often coincident with higher fracture intensity), whereas cooler colors indicate thinner seams (Figure 6). The integration of multiple attributes and AI has delineated sweet spots with high precision, aligning well with known well data.

*3.4. Micro-Structure Recognition and Fault Extraction*

Micro-faults, which have small displacements of less than ~20 m, often manifest as subtle, weak discontinuities in seismic sections. They can be difficult to detect using traditional interpretation methods or single-attribute analyses [23]. Identifying these small-scale structures is critical for CBM reservoirs because even minor faults can compartmentalize the reservoir or enhance local permeability through fracture development. In this study, we employed a combination of seismic attributes and an automated extraction algorithm to recognize and map these micro-structures.

Combined Attribute Approach: We specifically combined coherence, curvature, and ant-tracking attributes to automatically extract faults.

Coherence measures the similarity of seismic waveforms between adjacent traces and highlights reflector discontinuities, making it well-suited to identifying larger faults and stratigraphic edges [24]. We generated a coherence volume where low coherence values indicate potential fault planes or abrupt lateral changes.

Curvature attributes, which are derived from interpreted horizon surfaces, reveal the bending or flexure of layers. Areas of high curvature often correspond to folds or the damage zones of faults, thus they can indicate the presence and orientation of even subtle faults and fractures. We calculated both most-positive and most-negative curvature for key horizons, e.g., near the top of coal measures, to capture dome-and-basin like flexures or saddle-shaped warps.

Ant-tracking is an advanced volume-based fault detection algorithm that simulates the behavior of ants moving along discontinuities. It effectively enhances the continuity of fault-like lineaments by connecting discontinuous low-coherence or high-curvature features into coherent fault surfaces. We applied ant-tracking to a composite volume where coherence provided the primary discontinuity signal and curvature provided additional guidance on likely fault orientations. Key parameters for the ant-tracking process, such as the sensitivity threshold for what constitutes a discontinuity, the step size of the ant agents, and the number of iterations, were calibrated so that known major faults were readily detected and spurious noise-lineations were minimized. In our case, an ant-tracking parameter set with moderate sensitivity and 10 iterative cycles provided the best balance, successfully identifying subtle faults while filtering random noise.

Using this combined approach, the algorithm produced attribute volumes highlighting fault likelihood. We then extracted fault surfaces by thresholding the ant-tracking volume and converting connected high-likelihood voxels into planar fault patches. These automatically extracted faults were cross-validated and edited manually by an experienced interpreter to ensure geological plausibility and continuity. Multi-attribute cross-validation (checking that a candidate fault is consistently indicated by multiple attributes) and manual cleanup were essential steps to avoid false positives and to merge fragmented segments of the same fault.

The outcome is a high-resolution fault interpretation that includes numerous small faults which were previously unrecognized. Figure 7 shows a time-slice (plan view) of an interpreted curvature attribute overlaid with the ant-tracking result, delineating areas of intense fracturing and faulting. Bright, linear features on this map correspond to faults and fracture zones, and their orientations and densities can be observed. Figure 8 shows a representative seismic vertical section after interpretation, where several fine-scale faults have been picked out that were not part of the initial structural model. Coherence contributed to detecting the larger offsets, curvature helped in mapping subtle flexures associated with these faults, and ant-tracking connected the dots between broken seismic responses.

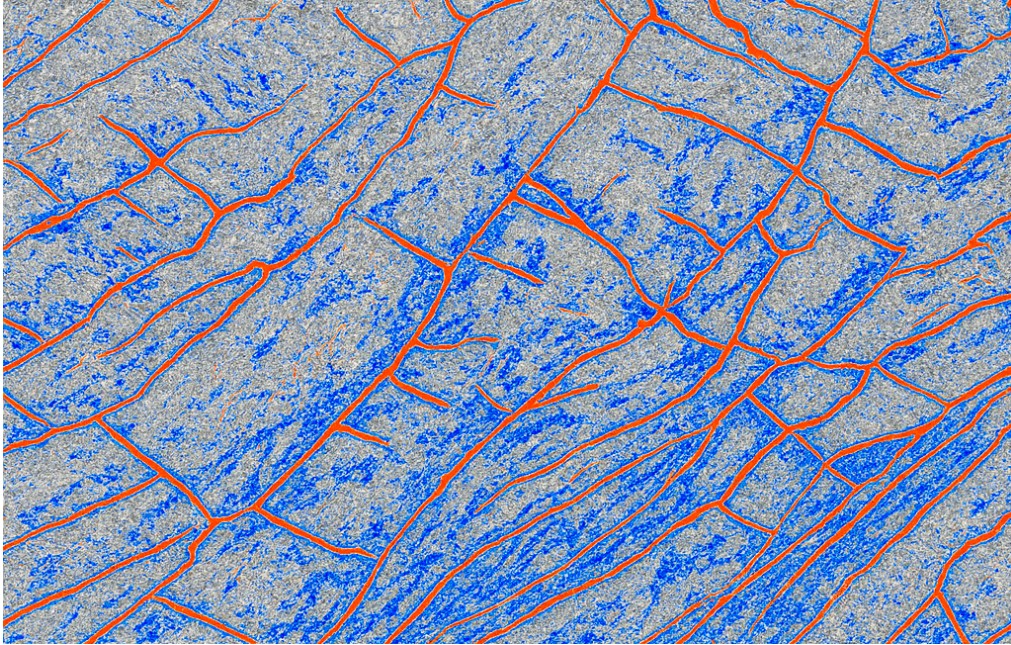

**Figure 7.** Plan view of interpreted curvature combined with ant-tracking body.

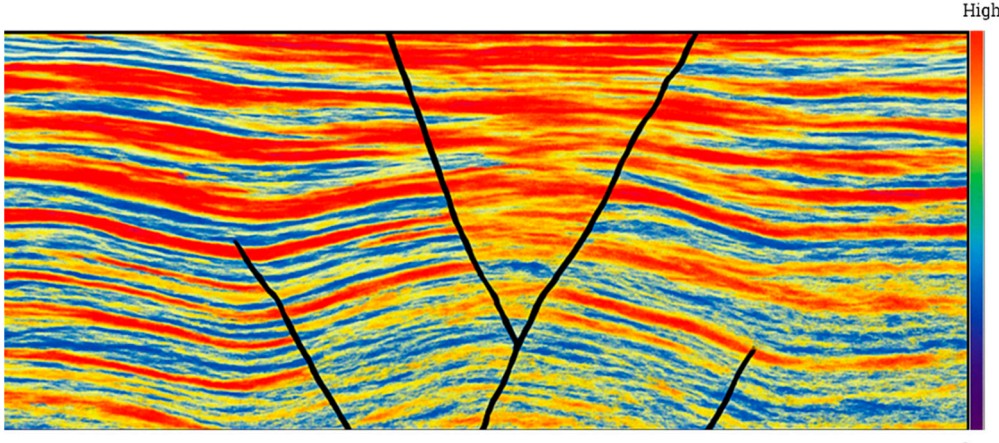

**Figure 8.** High-precision small fault seismic interpretation profiles.

### 3.5. Multi-Survey Data Integration

In parts of the study area, multiple 3D seismic surveys of different vintages and coverage were available. To maximize spatial coverage and interpretational consistency, we integrated five 3D seismic datasets into a unified volume. Merging multiple surveys requires careful handling of timing and amplitude discrepancies. We addressed the issue of temporal misalignment using a waveform matching technique. Overlapping areas of adjacent surveys were cross-correlated to determine the constant time shifts or subtle stretching/squeezing required for alignment, which is often referred to as closure correction. Small static shifts (on the order of 1–2 ms) were applied to remove any timing offsets between surveys. In addition, well tie analysis at several control wells was used to confirm that key horizon events from different surveys aligned within a few milliseconds.

To ensure amplitude consistency, we performed a robust scaling based on common histograms of amplitude in overlapping zones. Essentially, each survey's amplitude distribution was adjusted by a linear gain and bias to match a reference survey, ensuring that reflectors have comparable amplitudes across boundaries. This step prevents false discontinuities at survey seams that could be mistaken for geological features.

All datasets were then resampled to a common sample rate and merged. A unified interpretation framework was established: coal seam horizons were picked using a regional stratigraphic correlation that spans all surveys, and fault interpretations from Section 3.4 were extended across survey boundaries. This integration greatly improved the structural coherence, enabling us to trace features such as faults or channels in seismic data continuously, even if they crossed the original boundaries of individual surveys. It essentially laid the groundwork for integrated modeling by providing a single, high-quality seismic volume as the basis for subsequent steps.

### 3.6. Spatially Variable Velocity Modeling and Time-to-Depth Conversion

Although seismic interpretation is initially conducted in the time domain (i.e., two-way travel time), a depth-domain model is required for drilling and reservoir engineering purposes. Accurate time-to-depth conversion is thus essential, especially in structurally complex areas where velocity variations can lead to significant depth conversion errors [24]. We developed a refined 3D velocity model which takes into account spatial heterogeneity in velocities, for example, due to variations in rock composition, compaction levels or gas content.

We first established time–depth relationships at well locations using checkshot surveys and synthetic seismogram ties. The time pick of key seismic horizons, e.g., the top of GM seam, at each well was paired with its known depth from well logs to calculate average

interval velocities. Additionally, sonic log data provided continuous velocity information in wells, which we averaged over intervals corresponding to the main stratigraphic units. We also considered the stacking velocities from seismic processing as initial guides, though they were adjusted to well control for higher accuracy.

We imposed lateral constraints at the edges of the model and in areas of sparse well control using regional velocity trends derived from adjacent wells or regional studies. These constraints acted as boundary conditions, preventing unrealistic extrapolation and ensuring that the velocity model transitioned smoothly to known values outside the immediate study area. This is known as tying into an external velocity model.

We iteratively refined the velocity model by comparing the depth-converted seismic horizons to actual well marker depths. Any systematic discrepancies were used to adjust the velocity field locally. After a few iterations, the model achieved a high degree of accuracy: the depth conversion error for key marker horizons was reduced to within $\pm 0.5\%$ of depth, e.g., on a 500 m deep horizon, error $< \pm 2.5$ m. The final velocity model was therefore "spatially variable," capturing slower velocities in coal and shaly zones and faster velocities in sandstones or well-compacted strata, with smooth transitions.

Using this velocity model, we converted the interpreted time horizons, such as the Q, GM, and GL seam tops, from time to depth. The resulting depth structure maps honor well depths and account for the complex velocity variations across the field, thereby ensuring geological realism. This step is crucial for subsequent reservoir calculations and well trajectory planning, as even small depth errors can misplace a horizontal well within or outside a thin target seam.

*3.7. Three-Dimensional Structural Modeling*

With an accurate depth-converted interpretation of horizons and faults, we constructed a detailed 3D geological model of the coal reservoir. The model included the top and base surfaces of major coal seams and all significant fault planes, providing a geometric framework of the subsurface.

Each interpreted coal seam horizon (now in depth) was gridded at a fine resolution. Given the thin nature of many seams and the need to capture subtle undulations, we used a vertical layer increment of 1.0–1.5 m for model layering in the coal-bearing intervals. This fine layering allows the model to represent gentle dips and local rollovers that coarser layering would overlook. The horizon surfaces were constrained by well picks (drillers' depths for seam tops)—at those locations, the model surface was made to exactly honor the well depth. Between wells, the surfaces were guided by the seismic interpretation, which had already been quality-controlled with our velocity model.

Fault planes interpreted were input into the model as discrete surfaces cutting the stratigraphy. We ensured that fault positions and throws were consistent with both the seismic data and well correlations, e.g., where a fault offsets a seam between two wells, the throw in the model was calibrated to match that difference. Fault surfaces from coherence/ant-tracking were occasionally segmented; these were stitched together in the model if they aligned in three dimensions, resulting in continuous fault planes. Minor adjustments (on the order of a few meters) to fault geometries were sometimes made to ensure that fault and horizon intersections were geologically reasonable, e.g., avoiding small gaps or overlaps due to interpretation uncertainties.

The result is a 3D structural model that consists of a network of fault-bounded blocks and the geometry of coal seams within each block. This model provides a solid foundation for various applications: (a) volumetric calculations of gas resources (by combining seam thickness and area per block), (b) well planning (by visualizing horizontal well trajectories relative to seam undulations and faults), and (c) geomechanical modeling (by understand-

ing how faults and folds might influence stress distribution). The high level of detail—such as 1–1.5 m vertical resolution and inclusion of faults with throws as small as ~5 m—ensures that even subtle structural features influencing CBM extraction, e.g., gentle rollovers that a horizontal well might encounter, or small faults that could cause a pressure drop, are accounted for. In summary, the structural model captures the complex geology of the field at a resolution and accuracy that was not achievable with previous standard methods.

## 4. Results and Discussion (Case Study in the North Bowen Basin)

### 4.1. Comparison with Traditional Methods

Table 1 presents a direct comparison of our AI-based multi-attribute fusion with conventional interpretation methods in terms of thickness prediction error, fault recognition rate, and depth conversion accuracy. The results clearly demonstrate the advantages of the proposed framework. Specifically, the conventional RMS amplitude approach exhibits relatively large prediction errors (±2–3 m) and a low fault recognition rate (~65%), indicating its limited ability to resolve thin coal seams and subtle structural discontinuities. Multi-attribute regression improves the results to some extent, reducing thickness prediction error to ±1.5–2 m and increasing fault recognition accuracy to approximately 75%. However, the method still struggles to handle nonlinear attribute interactions and suffers from higher depth conversion errors (±6–8 m), which can lead to significant uncertainties in reservoir characterization and well placement.

**Table 1.** Performance comparison between traditional and AI-based methods.

| Method | Thickness Prediction Error | Fault Recognition Rate | Depth Conversion Error |
|---|---|---|---|
| Conventional RMS amplitude | ±2–3 m | ~65% | ±10–15 m |
| Multi-attribute regression | ±1.5–2 m | ~75% | ±6–8 m |
| AI-based multi-attribute fusion (this study) | ±1 m | 95% | ±2–3 m |

In contrast, the AI-based multi-attribute fusion method proposed in this study yields markedly better performance, achieving a thickness prediction error of only ±1 m, a fault recognition rate of 95%, and a depth conversion error as low as ±2–3 m. This indicates not only a substantial improvement in spatial resolution but also a more reliable mapping of fault and fracture systems. The enhanced performance stems from the ability of AI algorithms to capture nonlinear relationships among multiple seismic attributes, integrate geological constraints, and optimize prediction through iterative training. These improvements translate into more accurate reservoir characterization, reduced drilling risk, and more effective sweet-spot targeting for CBM development.

### 4.2. Post-Stack Seismic Data Processing

To improve imaging of coal seams and structural features, a workflow combining dynamic balancing and blue filtering was applied to the raw seismic data to mitigate the low S/N ratio and discontinuous reflections. The processed sections showed significantly better reflector continuity, making it easier to delineate thin seams and fault boundaries. The S/N ratio increased from 1.7 to 2.6, and the effective high-frequency bandwidth expanded from 25 Hz to 35 Hz. Furthermore, five 3D seismic datasets from various vintages were merged using precise stitching and closure correction techniques. Time mismatches of up to 2 ms between datasets were resolved to ensure a unified seismic interpretation framework, providing high-quality data for subsequent precision interpretation. Figure 9 shows the seismic sections before (left) and after (right) dynamic balancing and blue filtering, highlighting the significant improvements in continuity and resolution.

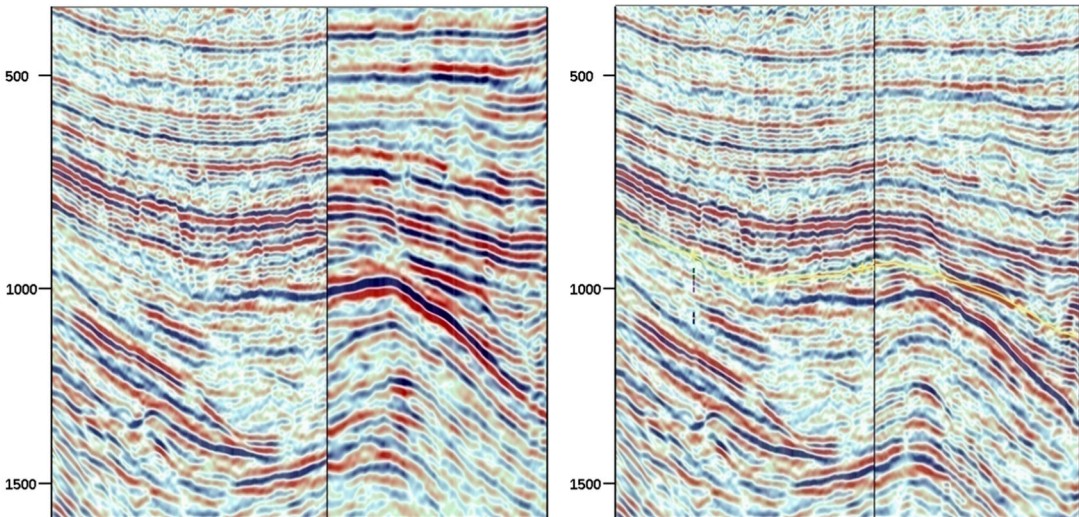

**Figure 9.** Seismic profiles before and after dynamic balancing and filtering treatment (**left**: before treatment, **right**: after treatment).

It is worth noting that these enhancements were achieved without introducing obvious artifacts or distortions, which speaks to the careful parameter choices, e.g., the window length for balancing and the spectral shape for filtering. We were cautious to avoid over-amplification of noise: for instance, the blue filter operator was tapered so that frequencies beyond 40 Hz, which in our data carried more noise than signal, were not excessively amplified. The net result is a seismic dataset that is much more interpretable. This forms a more robust basis for subsequent interpretation steps.

### 4.3. Stratigraphic Calibration and Structural Interpretation

Using the optimized seismic data and well logs from 293 wells, key coal seams, such as GM, Q, and GL, were identified via synthetic seismograms and waveform matching. Horizon calibration accuracy was controlled within $\pm 3$ ms. Multi-volume interpretation under a unified reference surface enabled continuous horizon tracking and fault surface identification. Through integrated analysis of coherence, curvature, and ant-tracking attributes, a total of 118 faults were interpreted, including 17 newly identified small-scale faults, mostly trending east–west. These micro-faults exhibit a dense spatial distribution and exert a significant control over coal seam geometry, forming gently inclined monoclines and broad folds. The resulting block segmentation pattern provides a structural basis for reservoir connectivity analysis and productivity prediction.

Geologically, the dense network of small normal faults is consistent with an extensional stress regime superimposed on the basin. Their predominantly east–west orientation implies a roughly north–south extension direction, which aligns with regional tectonic understanding for the late structural history of the North Bowen Basin. The fact that these faults have small throws but are numerous means they can significantly influence seam continuity without creating large offsets that would be obvious in older seismic data. Our detection of these faults is a direct result of the high-precision methods: the coherence attribute pinpointed discontinuities, curvature indicated flexure associated with fault drag, and ant-tracking connected pieces into coherent surfaces.

The impact of these fine-scale structures on the reservoir is important. The faults often delineate boundaries of compartments that could influence how gas pressure is distributed and how production might vary. For example, some of the small faults identified delineate the flanks of gentle folds (broad warps in the coal seams) that were also mapped; these folds can create local highs or lows in the seam elevation, potentially acting as gas migration paths

or barriers. We also observed that areas with clusters of small faults corresponded to zones of enhanced curvature on the seams, which were interpreted as areas of intense fracturing.

Overall, the structural interpretation paints a picture of the study area as a highly structured environment where both major and minor faults play a role. Traditional interpretation might have focused only on larger faults, where the throw exceeds 20 m, but our results demonstrate that including the smaller faults provides a far more complete structural model. This has direct practical implications: when planning horizontal wells, for instance, knowing about a 10 m fault in advance can be the difference between staying in the seam or exiting into the roof rock. Thus, the critical discussion point is that high-resolution seismic interpretation adds real value by capturing subtle geologic features that have tangible effects on development strategies.

### 4.4. Velocity Modeling and Geological Model Construction

Following the construction of the 3D velocity model and time-depth conversion, we evaluated the accuracy of the depth-converted seismic interpretation against well data and other controls. The refined velocity model proved to be highly accurate. Depth-converted horizon picks for key seams, such as Q, GM, GL, were compared to the actual drilled depths in numerous wells: the differences were generally within ±2–3 m, which is less than 0.5% of the depth (around 500–600 m in depth for these seams). This level of accuracy is a substantial improvement over initial conversion using a simplistic layer-cake or average velocity approach, which in early stages had errors of 10–20 m in some areas due to unaccounted lateral velocity changes.

The iterative calibration we performed, by adjusting the velocity model until the well ties converged, resulted in very tight control of structural elevation. In practical terms, this means that if a horizontal well is planned to stay 2 m below the top of the GM seam, our model can confidently predict where that top will be, within the margin of a couple of meters, even in areas away from well control. Moreover, the spatial variability captured in the model, such as slightly lower velocities in a coal-rich zone that cause a local pull-down in time and are now corrected in depth, ensures that structural closures or subtle dip changes are correctly positioned in depth.

Figure 10 presents a vertical velocity profile extracted along a line (L300) through the model, illustrating how velocities vary vertically and laterally. Coal-bearing layers are visible as slightly slower velocity zones (due to coal's low density and modulus) sandwiched between higher-velocity sandstones and siltstones. Near a fault zone in the line, one can see velocity perturbations where increased fracturing or localized lithology changes might have reduced velocities—a feature the model reflects by gently lowering velocities in that block, which is consistent with observed sonic log slowing. This highlights that our model is not just a smooth gradient but includes realistic heterogeneity.

The high-fidelity velocity model underpins the geological model's reliability. One critical discussion point is that depth conversion errors are a major source of uncertainty in conventional seismic interpretation; by reducing those errors to negligible levels (3% or less in horizon positioning, as noted), we significantly boost confidence in any depth-based analysis, such as volumetric estimations of gas in place or precise placement of wells. The process also underscores an interplay between geology and geophysics: our velocity model implicitly contains geological information, effectively acting as a bridge between the seismic two-way time domain and the real depth world.

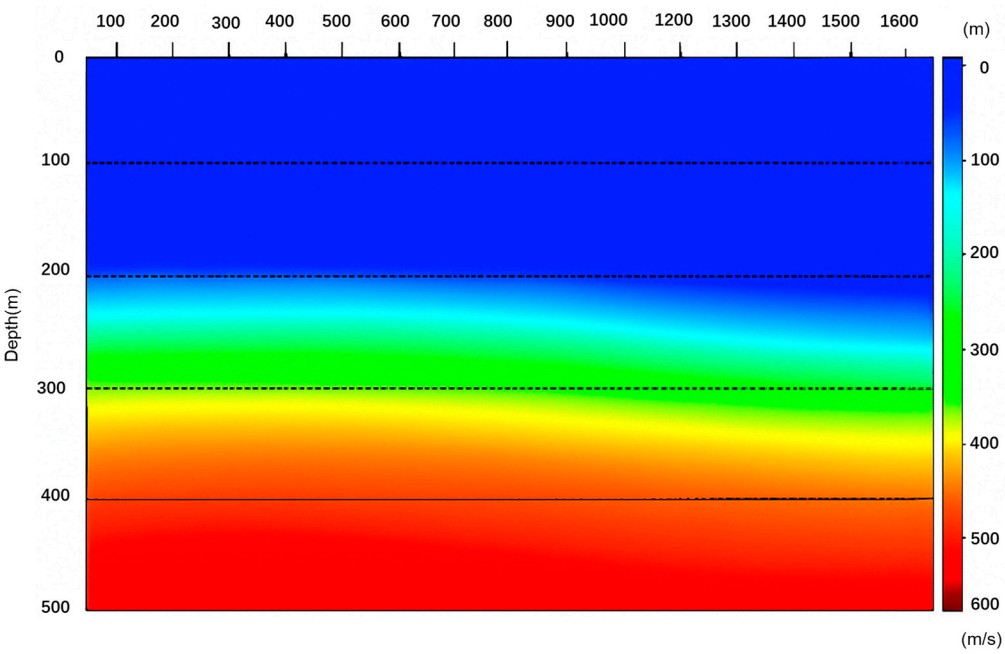

**Figure 10.** Velocity profile of the study area (L300).

*4.5. Seismic Attribute Analysis Results*

Following structural interpretation and model construction, seismic attributes were employed to predict reservoir parameters. A suite of attributes, including amplitude and instantaneous frequency, were calibrated against vertical well data using the Geophysics Pro (GeophPro) platform. Using the multi-attribute fusion approach with AI, we generated thickness maps for key seams. The results for the GM seam—our main target—show that its thickness typically ranges from 4 to 7 m across the field, consistent with well measurements. Importantly, the predictive model achieved errors generally within ±1 m when compared to blind well test points. This level of accuracy (roughly 15–25% of the seam thickness) is quite high for a seismic prediction of such a thin layer, indicating the effectiveness of integrating multiple attributes with a learning algorithm.

The maps reveal distinct lateral thickness anomalies. For example, certain northwest-trending corridor-like zones maintain GM seam thicknesses of 6–7 m (warm colors in Figure 10), whereas surrounding areas may thin to 3–4 m (cooler colors). These trends often correlate with structural features. One noteworthy observation is that areas of relatively greater thickness often occur on the flanks of gentle structural highs or along the hinges of broad folds. A likely geological explanation is that these are zones where the depositional or post-depositional conditions favored preservation of thicker coal, e.g., slower erosion or subsidence patterns that localized coal accumulation. Conversely, near some fault zones, the seam thickness drops off abruptly, which could be due to syndepositional fault activity causing differential accommodation or later erosion along uplifted sides of faults.

Attribute slice maps clearly reveal lateral trends in coal seam thickness and identify abrupt changes caused by fault activity. These results accurately characterize spatial variations in seam geometry and the role of fault-controlled gas accumulation, providing robust technical support for sweet spot delineation in CBM development. Figure 11 shows the predicted thickness distribution for the GM coal seam, derived from multi-attribute seismic analysis and displayed along the depth-converted horizon. Warmer colors denote thicker coal and cooler colors denote thinner coal. The map reveals thicker coal "corridors" and abrupt thickness changes across certain faults. These results highlight how structural deformation and depositional variability have led to non-uniform coal thickness, and they guide the identification of sweet spots where thickness and structural favorability coincide.

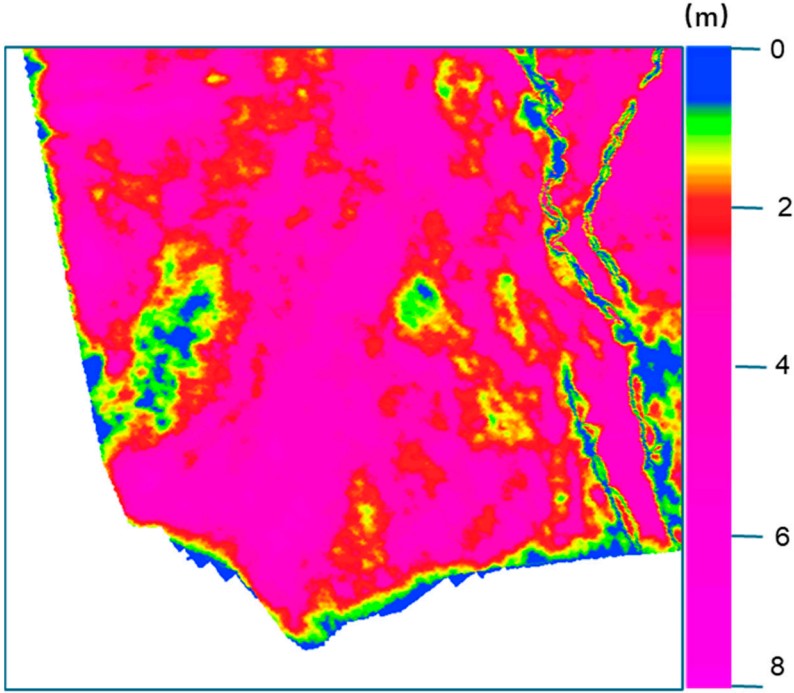

**Figure 11.** Distribution of GM coal seam (amplitude properties along the seam).

*4.6. Validation of Interpretation Accuracy Using Horizontal Well Trajectories*

A crucial part of our study was validating the interpretation results against independent data, namely the performance of horizontal wells in the field. Horizontal wells provide an excellent reality check because their trajectories sample the seam over long distances, and their drilling results and production outcomes reflect the true reservoir conditions. We conducted a comprehensive comparison between our seismic interpretation (structure and attributes) and five horizontal wells that were drilled in the study area targeting the GM seam. The validation was performed on several aspects:

Reservoir Encounter Accuracy: A quantitative assessment was conducted using the ratio of actual drilled coal seam length to the predicted length. Results show that reservoir encounter rates exceeded 95% in multiple key wells. This demonstrates the high spatial accuracy and continuity of the seismic attribute-based prediction model, which is sufficient to support horizontal well design and placement.

Wellbore Positioning Accuracy: Depth markers from logging-while-drilling data were used to identify the top and bottom of the coal encountered, and these were compared to the depths from our model at the same lateral positions. The average deviation between the predicted and actual seam positions was less than 3% of the true vertical depth. For a seam around 500 m deep, a 3% error corresponds to about 15 m, but in reality most points along the wells had differences under 5 m, with the occasional larger discrepancy in highly faulted areas. This level of precision confirms that the combination of detailed velocity modeling and structural interpretation has yielded a model where even in complex geological settings, the depth structure is highly reliable. Essentially, if our model says the seam is at 520 m TVD at some location, the well is indeed finding it roughly at that depth, give or take only a few meters.

Productivity Correlation and Attribute Validation: Finally, predicted reservoir parameters were cross-validated against actual production data. High-thickness zones (>4 m) coincided with regions of intense fracture development and were generally associated with high-production well segments. Within these combined 'thickness-fracture' sweet spots, initial daily gas production rates were 25–30% higher than those of traditional deployment

areas [25], indicating a strong positive correlation between predicted attributes and reservoir performance, and validating the practical value of the interpretation framework in resource evaluation and development planning.

To summarize, the validation phase reinforces the credibility of our approach. By achieving high reservoir encounter rates and positioning accuracy, we proved the structural model's fidelity. By demonstrating production uplifts in predicted sweet spots, we confirmed that the seismic attributes and AI predictions are capturing key reservoir quality differences. This level of validation is often missing in interpretation studies, and including it here strengthens the argument that the integrated high-precision seismic interpretation technology can directly translate into improved exploration and development success for CBM and similar resources.

## 5. Conclusions

This study addresses two key technical challenges in CBM exploration, i.e., the difficulty of identifying thin seams and the inaccuracy of structural interpretation. It proposes a high-precision seismic interpretation framework that integrates dynamic balancing, blue filtering, multi-attribute fusion and AI algorithms. This methodology has been successfully applied and validated in a CBM field in North Bowen Basin, Australia.

Through validation using multiple horizontal wells, including analyses of reservoir encounter accuracy, wellbore positioning precision, and production correlation, the results demonstrate the framework's high adaptability and engineering reliability in CBM reservoir identification and prediction.

Specifically, the combined application of dynamic amplitude balancing and blue spectral filtering substantially improved seismic data resolution and S/N ratio. This processing made thin seams and subtle faults much clearer on seismic sections, enabling interpretations that were not possible with the raw data alone.

Multi-attribute analysis, supported by AI, achieved coal seam thickness prediction errors within $\pm 1$ m for seams only a few meters thick. The approach revealed strong correlations between seismic attributes and geological properties, for instance, linking higher frequency/amplitude attributes to thicker coal sections. It also established, for the first time, a quantitative relationship between structural curvature (from seismic attributes) and fracture intensity. Together, these allow for accurate delineation of sweet spots (thicker, fractured coal zones) before drilling.

By utilizing coherence, curvature, and ant-tracking attributes, the framework automatically extracted a dense network of faults, including numerous small-scale faults that were previously undetected. Incorporating these into the geological model, along with a rigorously calibrated spatially variable velocity model, constrained time-to-depth conversion errors to within 0.5% in depth (horizon depth errors <3–5 m) and ensured high spatial accuracy in the 3D structural model of the reservoir.

The interpretation results were validated against horizontal well data, demonstrating >95% reservoir encounter rates and <3% depth deviation in well positioning. Furthermore, the predicted sweet spot zones corresponded closely with areas of 25–30% higher initial gas production rates compared to traditionally selected locations. This empirical validation confirms that the high-precision seismic interpretation framework provides reliable guidance for drilling and has a tangible positive impact on reservoir development outcomes.

In conclusion, the innovative integration of high-resolution seismic processing, attribute analysis, and AI-driven interpretation has markedly improved the precision and confidence of CBM reservoir characterization in the study area. The methodologies and insights from this work provide a strong geological and technical basis for optimizing CBM exploration and development strategies. By accurately predicting where the thickest,

most fractured, and gas-rich coal occurs, operators can target the most promising zones, reduce exploration risk, and enhance production. While demonstrated in a CBM context, the approach is general and can be applied to other geologically complex, thin-layered reservoirs. Ongoing and future work could extend this framework by incorporating time-lapse seismic to monitor production or by integrating geomechanical models to further understand fracture evolution. The success of this case study suggests that embracing high-precision seismic interpretation technology will be a key enabler in unlocking the full potential of unconventional reservoirs like CBM.

While the proposed framework demonstrates strong performance, it relies on dense well control and high-quality seismic data. Its effectiveness in areas with sparser data or lower S/N ratios remains to be tested. Future research should extend the approach to other basins, incorporate time-lapse seismic monitoring, and integrate geomechanical models to assess fracture evolution.

**Author Contributions:** Conceptualization, C.L.; Methodology, X.W.; Software, X.L.; Validation, L.D. and Z.D.; Investigation, X.W.; Data curation, L.F.; Writing—original draft, C.L.; Writing—review & editing, L.D. All authors have read and agreed to the published version of the manuscript.

**Funding:** The project was financially supported by the National Natural Science Foundation of China Key Project: "Evolution and Output Effect of Deep Coalbed Methane Formation" (No. 4230805), National Key Science and Technology Project: "Patterns of Coalbed Methane Enrichment and Exploration and Development Technologies in Key Basins of Overseas Exploration Areas" (No. 2025ZD1403904), PetroChina Science and Technology Program: "Research on Coal Bed Methane Enrichment Law and Development Mechanism" (No. 2024DJ23), and PetroChina Science and Technology Program: "Research on the Theory of Deep Coalbed Methane Deposition and Benefit Development Technology" (No. 2023ZZ18).

**Data Availability Statement:** The original contributions presented in this study are included in the article. Further inquiries can be directed to the corresponding author(s).

**Conflicts of Interest:** Author Xiuqin Lu was employed by the company Exploration and Development Research Institute of HuaBei Oilfield Company. Author Liyong Fan was employed by the company PetroChina Changqing Oilfield Company. The remaining authors declare that the research was conducted in the absence of any commercial or financial relationships that could be construed as a potential conflict of interest.

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
