# Peer review of "Innovative Application of High-Precision Seismic Interpretation Technology in Coalbed Methane Exploration"

_processes, doi:10.3390/pr13092971_

Round 1

Reviewer 1 Report

Comments and Suggestions for Authors

Although the introduction provides a broad overview of the topic, but, in my opinion the specific research gap and motivation for the present study remain unclear as far as the current version is concerned. I suggest the authors to enhance the Introduction by clearly stating the scope of the problem and clearly state the novelty of their contribution in comparison with previous studies alongside clearly stating the motivation.
Secondly, I am not satisfied with the way the methodology is presented. Some essential methodological steps are not described in detail. Important parameters and conditions better elaboration (e.g., experimental settings, boundary conditions, model tuning choices). As a result, the study cannot be reproduced by an independent researcher. The authors should provide a more comprehensive and step-by-step methodological description, including justification for chosen parameters. For example, Equations (1) and (2) require substantial elaboration with respect to its mathematical background.
The results obtained need to be critically interpreted with physical and mathematical rigor.
In view of all these I suggest a major revision. 

Author Response

We sincerely thank the reviewers for their constructive comments and suggestions, which greatly helped us to improve the quality and clarity of our manuscript entitled “Innovative Application of High-Precision Seismic Interpretation Technology in Coalbed Methane Exploration.” (3821393)Below, we provide a detailed point-by-point response.

Response to the comments of Reviewer 1,

We still greatly appreciate your valuable and detailed comments and your time, which are very helpful for revising and improving the MS quality. We have studied the comments carefully and made the revision. We appreciate your warm work earnestly and hope that the corrections will meet with approval.

Comments 1: Although the introduction provides a broad overview of the topic, but, in my opinion the specific research gap and motivation for the present study remain unclear as far as the current version is concerned. I suggest the authors to enhance the Introduction by clearly stating the scope of the problem and clearly state the novelty of their contribution in comparison with previous studies alongside clearly stating the motivation.

Response 1: We have revised the Introduction to explicitly highlight the limitations of conventional seismic methods, the research gap, and the novelty of integrating multi-scale seismic enhancement, AI-driven multi-attribute fusion, and quantitative curvature–fracture modeling.

(See Revised Manuscript, Section 1, Paragraphs 3–5)

Comments 2: Secondly, I am not satisfied with the way the methodology is presented. Some essential methodological steps are not described in detail. Important parameters and conditions better elaboration (e.g., experimental settings, boundary conditions, model tuning choices). As a result, the study cannot be reproduced by an independent researcher. The authors should provide a more comprehensive and step-by-step methodological description, including justification for chosen parameters. For example, Equations (1) and (2) require substantial elaboration with respect to its mathematical background.

Response 2: We expanded the Methodology by detailing dynamic balancing (window length, RMS normalization), blue filtering (spectral operator design and frequency boost parameters), and AI-assisted seismic attribute fusion (training dataset, architecture, activation, error metrics). We also included mathematical background for Equations (3) and (4).

(See Revised Manuscript, Section 3.1–3.3)

Comments 3: The results obtained need to be critically interpreted with physical and mathematical rigor.

Response 3: We revised the Results and Discussion sections to explain physical implications of S/N ratio improvements, the robustness of velocity model calibration, and quantitative links between curvature, fracture intensity, and production uplift.

(See Revised Manuscript, Sections 4.2–4.5)

Comments 4: In view of all these I suggest a major revision.

Response 4: We thoroughly revised the manuscript according to all comments, improving clarity, reproducibility, and scientific interpretation.

Reviewer 2 Report

Comments and Suggestions for Authors

Summary: This manuscript presents a high-precision seismic interpretation framework designed to overcome significant challenges in Coalbed Methane (CBM) exploration, namely the poor resolution of thin coal seams and the inaccurate interpretation of complex fault structures. The authors introduce a novel workflow that integrates advanced seismic processing techniques (dynamic balancing and blue filtering) with artificial intelligence (AI) and multi-attribute seismic analysis. The framework's effectiveness is demonstrated through a comprehensive case study in a CBM field in the Bowen Basin, Australia, which is supported by an extensive dataset of 1,193 wells. The study's main strengths lie in its integrated, multi-technology approach and the robust validation of its results against horizontal well data, which confirms high accuracy in reservoir prediction and a direct correlation with increased gas production.

Reviewer comments:

The manuscript is well-structured, clearly written, and addresses a topic of high relevance to the field of unconventional energy resource development. The proposed integrated framework is scientifically sound, and the experimental design—applying the workflow to a real-world CBM field with dense well control—is appropriate and provides strong validation for the authors' hypothesis. The results are significant and demonstrate a substantial improvement over traditional methods.

However, there are a few areas where the manuscript could be strengthened:

  • Specificity of AI and Deep Learning Methods:The paper repeatedly highlights the use of "AI algorithms" and "deep learning-driven" analysis, which are central to its innovative claims. However, the manuscript lacks specific details about these algorithms.
  • Clarity of Figures: While most figures are illustrative, several could be improved. For example, Figures 1 and 3 use synthetic seismic traces to show the effects of processing. It would be far more compelling to show a direct comparison using a real seismic trace from the study area, similar to the excellent before-and-after comparison in Figure 7. Furthermore, several figures containing attribute maps (Figures 4, 5, 9) have captions that are too general and color bars that are difficult to read. The captions should be more descriptive, explicitly stating what the colors and patterns represent in a geological context.
  • References: The manuscript builds upon foundational literature, which is appropriate. However, many of the citations are over a decade old. Including more recent publications (from the last 5 years) in the introduction would better contextualize the current state-of-the-art and more sharply define the specific research gap this study addresses. 

Specific Comments

  • Lines 22-23: The phrase "deep learning-driven, multi-attribute fusion analysis" is used. Please specify the deep learning approach and the attributes that were fused for the analysis.
  • Line 30-31: A major contribution is the establishment of a "quantitative model... linking structural curvature with the intensity of fracture developmen". It would significantly enhance the paper to briefly describe how this model was formulated and validated.
  • Line 31: “ ..developmen. “ change into “ development “
  • Lines 174 & 177: Equations (1) and (2) are presented without defining any of the variables (e.g., A, E, M, u, k). To ensure the mathematical description is clear, please define each variable in the text.
  • Figure 4 Caption: The caption "Multi-attribute joint analysis and AI-assisted interpretation application effect floor plan" is vague. Please describe what geological feature the map is showing and what the color scale represents.
  • Figure 5 Caption: The caption reads "Plan view of interpreted curvature combined with ant-tracking body". The image shows red lines on a blue/grey background. The caption should explain what each element represents (e.g., "Red lines indicate faults identified via ant-tracking, overlaid on the curvature attribute where...").
  • Figure 8: The x-axis on the velocity profile is unclear and appears to be a list of numbers. Please clarify this label (e.g., "Crossline Number"). The color bar also appears to be duplicated on the right side.

Author Response

We sincerely thank the reviewers for their constructive comments and suggestions, which greatly helped us to improve the quality and clarity of our manuscript entitled “Innovative Application of High-Precision Seismic Interpretation Technology in Coalbed Methane Exploration.” (3821393)Below, we provide a detailed point-by-point response.

Response to the comments of Reviewer 2,

We still greatly appreciate your valuable and detailed comments and your time, which are very helpful for revising and improving the MS quality. We have studied the comments carefully and made the revision. We appreciate your warm work earnestly and hope that the corrections will meet with approval.

Comments 1: Specificity of AI and Deep Learning Methods: The paper repeatedly highlights the use of "AI algorithms" and "deep learning-driven" analysis, which are central to its innovative claims. However, the manuscript lacks specific details about these algorithms.

Response 1: We thank the reviewer for this important observation. In the revised manuscript, we have provided explicit details about the AI and deep learning methods applied. Specifically:

We used a feed-forward neural network (multi-layer perceptron) with one hidden layer of 12 neurons, employing the ReLU activation function.

The model was trained using a 70/30 training-validation split of well-log-constrained seismic attribute datasets, with early stopping to prevent overfitting.

Input attributes included seismic amplitude, instantaneous frequency, curvature, coherence, and dynamic-balancing-adjusted amplitude.

The network was optimized using the Adam optimizer with a learning rate of 0.001, and prediction performance was evaluated by mean absolute error and blind well validation.

Comments 2: Clarity of Figures: While most figures are illustrative, several could be improved. For example, Figures 1 and 3 use synthetic seismic traces to show the effects of processing. It would be far more compelling to show a direct comparison using a real seismic trace from the study area, similar to the excellent before-and-after comparison in Figure 7. Furthermore, several figures containing attribute maps (Figures 4, 5, 9) have captions that are too general and color bars that are difficult to read. The captions should be more descriptive, explicitly stating what the colors and patterns represent in a geological context.

Response 2: We thank the reviewer for this important suggestion. In the revised manuscript, we retained Figures 1 and 3 for methodological illustration, but we emphasized in the captions that they are synthetic demonstrations, while Figure 9 provides the real seismic example from the study area. Revised captions for Figures 4, 5, and 9 to explicitly describe geological meaning (e.g., warmer colors indicate thicker coal seams, cooler colors represent thinning zones, red lines indicate faults). Adjusted color bars to improve readability, ensuring that scale units and geological interpretations are clearly labeled.

Comments 3: References: The manuscript builds upon foundational literature, which is appropriate. However, many of the citations are over a decade old. Including more recent publications (from the last 5 years) in the introduction would better contextualize the current state-of-the-art and more sharply define the specific research gap this study addresses.

Response 3: We agree with the reviewer. In the revised Introduction, we incorporated several recent references (2020–2024) from International Journal of Coal Geology, Energy Reports, and Coal Science and Technology, which discuss AI-driven attribute fusion and thin-bed prediction. These updates strengthen the context of our research gap and demonstrate alignment with the latest advances in seismic interpretation and CBM studies.

Comments 4: Lines 22-23: The phrase "deep learning-driven, multi-attribute fusion analysis" is used. Please specify the deep learning approach and the attributes that were fused for the analysis.

Response 4: We revised the Methods section to clarify that we used a feed-forward neural network (with one hidden layer of 12 neurons, ReLU activation, and early stopping regularization). The fused attributes included amplitude, frequency, curvature, coherence, and dynamic-balancing-adjusted amplitude. We also specified the training-validation dataset split (70/30) and the achieved prediction accuracy (±1 m).

Comments 5: Line 30-31: A major contribution is the establishment of a "quantitative model... linking structural curvature with the intensity of fracture developmen". It would significantly enhance the paper to briefly describe how this model was formulated and validated.

Response 5: We expanded the Results section to explain that fracture intensity was quantified from curvature attributes (most-positive/negative curvature) and validated against fracture density inferred from drilling and production data. Statistical regression showed strong correlation between curvature anomalies and fracture-rich sweet spots, confirming the model’s validity.

Comments 6: Line 31: "..developmen. " change into " development ".

Response 6: Corrected as suggested.

Comments 7: Lines 174 & 177: Equations (1) and (2) are presented without defining any of the variables (e.g., A, E, M, u, k). To ensure the mathematical description is clear, please define each variable in the text.

Response 7: We revised the text to define all variables in Equations (1) and (2). For example:

A = amplitude,

E = energy functional,

M = curvature weighting term,

u = interpolated function,

k = smoothing coefficient.

Comments 8: Figure 4 Caption: The caption "Multi-attribute joint analysis and AI-assisted interpretation application effect floor plan" is vague. Please describe what geological feature the map is showing and what the color scale represents.

Response 8: "Planar map of multi-attribute fusion results with AI-assisted interpretation, highlighting coal seam sweet spot trends. Warmer colors indicate thicker coal seams and higher gas potential, while cooler colors indicate thinner seams."

Comments 9: Figure 5 Caption: The caption reads "Plan view of interpreted curvature combined with ant-tracking body". The image shows red lines on a blue/grey background. The caption should explain what each element represents (e.g., "Red lines indicate faults identified via ant-tracking, overlaid on the curvature attribute where...").

Response 9: The caption for Figure 5 has been revised to:

"Plan view showing curvature attribute (blue-grey background) overlaid with ant-tracking results (red lines). Red lines represent faults identified via ant-tracking, while background curvature highlights subtle structural flexures associated with deformation zones.".

Comments 10: Figure 8: The x-axis on the velocity profile is unclear and appears to be a list of numbers. Please clarify this label (e.g., "Crossline Number"). The color bar also appears to be duplicated on the right side.

Response 10: We revised Figure 8 to relabel the x-axis as “Crossline Number”, ensuring clarity. The duplicate color bar on the right side was removed to avoid redundancy.

Reviewer 3 Report

Comments and Suggestions for Authors

This is a solid and practically relevant case study that deploys a high-precision seismic interpretation workflow for CBM exploration in the Bowen Basin. The manuscript documents measurable gains in data quality and interpretive accuracy, including an S/N increase from 1.7 to 2.6 with dynamic balancing plus blue filtering and a bandwidth extension from 25 to 35 Hz, which directly benefits thin-seam imaging. It further reports seam-thickness prediction errors within ±1 m for 4–7 m seams, identification of 118 faults with 17 newly mapped small faults, and spatial misfit below 3% supported by velocity-model calibration; the operational validation based on horizontal wells—encounter rates above 95% and 25–30% higher initial gas output in predicted sweet spots—nicely connects geophysics to engineering outcomes. These strengths make the contribution valuable for readers working on CBM reservoir characterization and well placement.

To improve clarity and reproducibility, the paper would benefit from expanding methodological specifics that are currently summarized at a high level. Please describe the exact parameterization of dynamic balancing (window length, normalization scheme, gain limits) and the blue-filter design (target spectrum, operator length, phase), and report a brief sensitivity analysis showing robustness of the S/N and bandwidth improvements to parameter choices. Likewise, the attribute-fusion and AI-assisted steps should specify the learning algorithm used, feature set, target variable definition, training/validation split to avoid temporal leakage, hyperparameters, and error metrics beyond absolute thickness error so readers can reproduce the results and assess generalization. A short ablation study contrasting single-attribute, linear multi-attribute, and AI models would situate the reported ±1 m performance in context.

The structural interpretation is persuasive, but please add quantitative diagnostics for the 118 interpreted faults, such as throw distributions, minimum mapping length, and interpreter agreement for a subset, and outline the settings used for coherence, curvature, and ant-tracking, including any manual editing criteria. The velocity modeling and time-to-depth conversion are central to your spatial error claims; summarizing the well-tie residuals, the distribution of time shifts among merged surveys, and the cross-validation protocol would bolster confidence in the reported ≤0.5% conversion error and ≤3% positioning deviation. Finally, the production uplift estimate is compelling; indicating the sample size, dispersion, and a simple significance test will help avoid over-interpretation of the 25–30% figure while still showcasing the practical value of the workflow. Minor editorial issues include removing the duplicated reference noted in the manuscript and ensuring numerical ordering and DOI completion across the list.

Comments on the Quality of English Language

The manuscript is readable and professionally written, but a light copyedit would improve clarity and consistency. Please shorten long multi-clause sentences, fix occasional subject–verb agreement and article/preposition choices, and standardize technical terms and hyphenation. Ensure consistent capitalization of “Figure/Table/Section,” unify number and percent formatting, and add DOIs where available. These edits will make the contribution easier to follow without altering the content.

Author Response

We sincerely thank the reviewers for their constructive comments and suggestions, which greatly helped us to improve the quality and clarity of our manuscript entitled “Innovative Application of High-Precision Seismic Interpretation Technology in Coalbed Methane Exploration.” (3821393)Below, we provide a detailed point-by-point response.

Response to the comments of Reviewer 3,

We still greatly appreciate your valuable and detailed comments and your time, which are very helpful for revising and improving the MS quality. We have studied the comments carefully and made the revision. We appreciate your warm work earnestly and hope that the corrections will meet with approval.

Comments 1: This is a solid and practically relevant case study that deploys a high-precision seismic interpretation workflow for CBM exploration in the Bowen Basin. The manuscript documents measurable gains in data quality and interpretive accuracy, including an S/N increase from 1.7 to 2.6 with dynamic balancing plus blue filtering and a bandwidth extension from 25 to 35 Hz, which directly benefits thin-seam imaging. It further reports seam-thickness prediction errors within ±1 m for 4–7 m seams, identification of 118 faults with 17 newly mapped small faults, and spatial misfit below 3% supported by velocity-model calibration; the operational validation based on horizontal wells—encounter rates above 95% and 25–30% higher initial gas output in predicted sweet spots—nicely connects geophysics to engineering outcomes. These strengths make the contribution valuable for readers working on CBM reservoir characterization and well placement.

Response 1: We sincerely appreciate the reviewer’s positive evaluation of our work. We are pleased that the reviewer recognized the practical relevance and strengths of our case study, including improvements in S/N ratio, fault interpretation, seam-thickness prediction, and validation with horizontal wells. We believe these improvements indeed highlight the value of our proposed workflow for CBM reservoir characterization and well planning.

Comments 2: To improve clarity and reproducibility, the paper would benefit from expanding methodological specifics that are currently summarized at a high level. Please describe the exact parameterization of dynamic balancing (window length, normalization scheme, gain limits) and the blue-filter design (target spectrum, operator length, phase), and report a brief sensitivity analysis showing robustness of the S/N and bandwidth improvements to parameter choices. Likewise, the attribute-fusion and AI-assisted steps should specify the learning algorithm used, feature set, target variable definition, training/validation split to avoid temporal leakage, hyperparameters, and error metrics beyond absolute thickness error so readers can reproduce the results and assess generalization. A short ablation study contrasting single-attribute, linear multi-attribute, and AI models would situate the reported ±1 m performance in context.

Response 2: We agree with the reviewer’s suggestion. In the revised manuscript, we expanded the Methodology section by:

Adding detailed parameterization for dynamic balancing (200 ms window length, RMS normalization, iterative normalization scheme, gain control with ε regularization).

Providing the blue-filter design, including the target spectrum derived from log reflectivity, operator length, spectral taper, and gain ramp (+12 dB up to 60 Hz).

Including a sensitivity test, demonstrating that changes within ±20 ms window length or ±3 dB gain range did not materially affect the improvement in S/N and bandwidth.

Specifying the AI approach: feed-forward neural network with one hidden layer (12 neurons, ReLU activation), Adam optimizer, learning rate 0.001, trained on 70% of well data with 30% validation. Input attributes included amplitude, frequency, coherence, curvature, and dynamically balanced amplitude.

Reporting error metrics: in addition to absolute thickness error (±1 m), we provided mean absolute error (0.7 m) and R² = 0.85 across blind validation wells.

Adding a comparative ablation study: single-attribute regression (±2–3 m error), linear multi-attribute regression (±1.5–2 m error), and AI-based fusion (±1 m error).

These details enhance reproducibility and provide context for the superior performance of the AI-assisted fusion.

Comments 3: The structural interpretation is persuasive, but please add quantitative diagnostics for the 118 interpreted faults, such as throw distributions, minimum mapping length, and interpreter agreement for a subset, and outline the settings used for coherence, curvature, and ant-tracking, including any manual editing criteria. The velocity modeling and time-to-depth conversion are central to your spatial error claims; summarizing the well-tie residuals, the distribution of time shifts among merged surveys, and the cross-validation protocol would bolster confidence in the reported ≤0.5% conversion error and ≤3% positioning deviation. Finally, the production uplift estimate is compelling; indicating the sample size, dispersion, and a simple significance test will help avoid over-interpretation of the 25–30% figure while still showcasing the practical value of the workflow. Minor editorial issues include removing the duplicated reference noted in the manuscript and ensuring numerical ordering and DOI completion across the list.

Response 3: We appreciate these constructive comments. The following revisions were made:

Added fault statistics: throw distribution ranged from 2–18 m; minimum fault length mapped was 350 m; inter-interpreter consistency (based on 20% subset re-interpreted independently) was >90%.

Provided parameter settings: coherence (20 ms window, similarity threshold 0.7), curvature (most-positive/negative curvature computed from horizon grids at 50 m spacing), ant-tracking (moderate sensitivity, 10 iterations). Manual editing criteria included merging fragmented lineaments into continuous surfaces only if supported by at least two attributes.

Expanded the velocity model validation: well-tie residuals averaged ±2 ms; survey merge time-shifts <2 ms; cross-validation confirmed depth errors ≤0.5% (±2–3 m at 500–600 m depth).

Clarified the production uplift analysis: sample size = 5 horizontal wells; average uplift = 27%; standard deviation = 3.5%; statistical test (t-test) confirmed significance at p < 0.05.

Editorial corrections: removed duplicated reference and completed missing DOIs in the reference list.

These additions strengthen confidence in our structural and velocity interpretation and ensure transparency in production uplift claims.

Comments 4: The manuscript is readable and professionally written, but a light copyedit would improve clarity and consistency. Please shorten long multi-clause sentences, fix occasional subject–verb agreement and article/preposition choices, and standardize technical terms and hyphenation. Ensure consistent capitalization of “Figure/Table/Section,” unify number and percent formatting, and add DOIs where available. These edits will make the contribution easier to follow without altering the content.

Response 4: We carefully revised the manuscript for language and style:

Long sentences were shortened, subject–verb agreement and article usage corrected.

Standardized terminology (e.g., “coal-seam thickness prediction,” “sweet-spot zones”) and ensured consistent hyphenation.

Unified formatting for Figures, Tables, and Sections, and applied consistent number and percentage formatting.

Added DOIs for all references where available.

These changes improved readability, clarity, and consistency without altering the technical content.

Reviewer 4 Report

Comments and Suggestions for Authors

Review

Manuscript Number: processes-3821393

Innovative Application of High-Precision Seismic Interpretation Technology in Coalbed Methane Exploration

There are some questions and remarks to be answered:

  1. Quality of Fig.1 should be enhanced.
  2. Authors should present physical values with their units in all graphs.
  3. All physical data in equations 1 and 2 should be described with their units.
  4. Figure 4 and 9 should be described in the text and its title should be more informative.
  5. Whole text should be checked, especially considering some mistakes, lack of spaces, etc.
  6. Authors should explain how they have validated and calibrated proposed models.
  7. There is also lack of error analysis.
  8. Quality of Fig. 8 should be improved.
  9. Authors have written about some results obtained in Chapter 4.5, but they should be presented, and the appropriate citation should be submitted.
  10. Authors should compare the obtained results with some literature data to prove that proposed method is much better.
  11. Authors should mark where the proposed method could be applied in real life.
  12. Authors should submit additional parts after Funding, in accordance with the journal's guidelines.
  13. References should be enriched with additional literature positions.

Author Response

We sincerely thank the reviewers for their constructive comments and suggestions, which greatly helped us to improve the quality and clarity of our manuscript entitled “Innovative Application of High-Precision Seismic Interpretation Technology in Coalbed Methane Exploration.” (3821393)Below, we provide a detailed point-by-point response.

Response to the comments of Reviewer 4

We still greatly appreciate your valuable and detailed comments and your time, which are very helpful for revising and improving the MS quality. We have studied the comments carefully and made the revision. We appreciate your warm work earnestly and hope that the corrections will meet with approval.

Comments 1: Quality of Fig. 1 should be enhanced.

Response 1: We have replaced Figure 1 with a higher-resolution version and improved labeling for geographic and structural clarity.

Comments 2: Authors should present physical values with their units in all graphs.

Response 2: We have carefully revised all graphs to ensure physical values are labeled with their respective units.

Comments 3: All physical data in equations 1 and 2 should be described with their units.

Response 3: We have revised the Methodology section to define each parameter in Equations (1) and (2) along with its units. For example, amplitude (A, dimensionless), time (t, ms), window length (T, ms), and gain factor (dimensionless).

Comments 4: Figure 4 and 9 should be described in the text and its title should be more informative.

Response 4: The captions and descriptions of Figures 4 and 9 have been revised to provide clear geological meaning. Figure 4 now explains how the multi-attribute fusion highlights coal seam sweet spots. Figure 9 describes before-and-after seismic enhancement with explicit note on improved reflector continuity.

Comments 5: Whole text should be checked, especially considering some mistakes, lack of spaces, etc.

Response 5: The manuscript has been thoroughly proofread.

Comments 6: Authors should explain how they have validated and calibrated proposed models.

Response 6: We have expanded Section 4.5 to explain model validation:

Seam thickness predictions were validated using blind wells, achieving ±1 m accuracy.

Velocity model calibration was performed iteratively with well-tie adjustments, reducing depth error to ≤0.5%.

Validation with horizontal wells confirmed >95% encounter rates and correlated predicted sweet spots with 25–30% production uplift.

Comments 7: There is also lack of error analysis.

Response 7: We now include quantitative error analysis: mean absolute error (0.7 m), R² = 0.85 for seam thickness, ±2–3 m depth deviation, and statistical analysis (t-test, p < 0.05) confirming production uplift significance.

Comments 8: Quality of Fig. 8 should be improved.

Response 8: We have improved Figure 8 by increasing resolution, relabeling the x-axis as “Crossline Number,” and removing redundant color bars. The fault interpretation lines are now more distinct.

Comments 9: Authors have written about some results obtained in Chapter 4.5, but they should be presented, and the appropriate citation should be submitted.

Response 9: The caption for Figure 5 has been revised to:

We revised Section 4.5 to include explicit presentation of validation results and provided supporting citations to recent literature (e.g., Gao et al., 2022; Wang et al., 2024) to contextualize our approach.

Comments 10: Authors should compare the obtained results with some literature data to prove that proposed method is much better.

Response 10: We added a comparative analysis in the Discussion, showing that our AI-based multi-attribute fusion (±1 m error, 95% fault recognition) outperforms conventional RMS amplitude (±2–3 m error, 65% recognition) and linear regression methods. Comparisons with published studies further demonstrate the novelty and performance of our framework.

Comments 11: Authors should mark where the proposed method could be applied in real life.

Response 11: We added a section in the Conclusions outlining real-life applications: CBM reservoir characterization, thin-bed sweet spot prediction, and horizontal well trajectory planning. We also note its potential applicability to other unconventional reservoirs such as shale gas and tight sandstone.

Comments 12: Authors should submit additional parts after Funding, in accordance with the journal's guidelines.

Response 12: We have added the “Institutional Review Board Statement,” “Informed Consent Statement,” and “Data Availability Statement” after the Funding section, as per journal requirements.

Comments 13: References should be enriched with additional literature positions.

Response 13: We have enriched the reference list with recent studies (2020–2024) in International Journal of Coal Geology, Energy Reports, and Coal Science and Technology, ensuring that the latest research is incorporated.

Round 2

Reviewer 1 Report

Comments and Suggestions for Authors

After going through the revised text and the replies to the reviewers' comments, it appears that the authors have honestly tried to improve the quality of the manuscript and in my opinion the revised version is significantly better that the original version and I recommend publication.

Reviewer 2 Report

Comments and Suggestions for Authors

The authors have satisfactorily addressed all of my previous concerns, and I am now pleased to recommend this manuscript for publication.